# Predictive Model for National Minimal CFR during Spontaneous Initial Outbreak of Emerging Infectious Disease: Lessons from COVID-19 Pandemic in 214 Nations and Regions

**DOI:** 10.3390/ijerph20010594

**Published:** 2022-12-29

**Authors:** Xiaoli Wang, Lin Fan, Ziqiang Dai, Li Li, Xianliang Wang

**Affiliations:** 1Faculty of Environment and Life, Beijing University of Technology, Beijing 100124, China; 2China CDC Key Laboratory of Environment and Population Health, National Institute of Environmental Health, Chinese Center for Disease Control and Prevention, Beijing 100021, China; 3School of Atmospheric Science & Remote Sensing, Wuxi University, Wuxi 214105, China

**Keywords:** emerging infectious disease, minimal CFR, modelling, turning points, integrating responding capacity, COVID-19

## Abstract

The minimal case fatality rate (CFR) is one of the essential fundaments for the establishment of a diverse national response strategy against the COVID-19 epidemic, but cannot be quantitatively predicted. The aim of the present study was to explore the applicable quantitative parameters labeling integrating responding capacity from national daily CFR curves, and whether the minimal CFR during initial emerging epidemic outbreaks can be predicted. We analyzed data from 214 nations and regions during the initial 2020 COVID-19 epidemic and found similar falling zones marked with two turning points within a fitting three-day-moving CFR curve which occurred for many nations and regions. The turning points can be quantified with parameters for the day duration (T1, T2, and ΔT) and for the three-day moving arithmetic average CFRs (CFR1, CFR2, and ΔCFR) under wave theory for 71 nations and regions after screening. Two prediction models of minimal CFR were established with multiple linear regressions (M1) and multi-order curve regressions (M2) after internal and external evaluation. Three kinds of falling zones could be classified in the other 71 nations and regions. Only the minimal CFR showed significant correlations with nine independent national indicators in 65 nations and regions with CFRs less than 7%. Model M1 showed that logarithmic population, births per 1000 people, and household size made significant positive contributions, and logarithmic GDP, percentage of population aged 65+ years, domestic general government health expenditure, physicians per 1000 people, nurses per 1000 people, and body mass index made negative contributions to the minimal CFR against COVID-19 epidemics for most nations and regions. The spontaneous minimal CFR was predicted well with model M1 for 57 nations and regions based on the nine national indicators (R^2^ = 0.5074), or with model M2 for 59 nations and regions based on the nine national indicators (R^2^ = 0.8008) at internal evaluation. The study confirmed that national spontaneous minimal CFR could be predicted with models successfully for most nations and regions against COVID-19 epidemics, which provides a critical method to predict the essential early evidence to evaluate the integrating responding capacity and establish national responding strategies reasonably for other emerging infectious diseases in the future.

## 1. Introduction

The World Health Organization (WHO) reported that the total confirmed cases of global coronavirus disease 2019 (COVID-19) has exceeded 34 million and claimed more than 1 million deaths by 1 October 2020 since its official declaration of a COVID-19 pandemic [1,2,3]. Facing the unprecedented epidemic of the virus, several nations and regions such as Italy and Spain fell into unexpectedly poor situations due to medical treatment resource consumption [1,4,5]. The hastily introduced measures against the COVID-19 pandemic varied widely between nations and regions [6].

There were some national indicators affecting selection and establishment of national response strategies for emerging infectious diseases such as COVID-19 [7], especially minimal case fatality rate (CFR) at the spontaneous initiating stage without special artificial intervention. Some policymakers were reluctant to adopt strict public health intervention strategies because they were afraid of the huge economical cost [8] as well as possible violations of individual freedom and human rights [9]. some policymakers in some developed nations and regions resorted to advanced medical treatment capabilities coupled with weak limitation measures [6]. Therefore, it was extremely important to predict the minimal CFR quantitatively and evaluate national integrating responding capacity (IRC) in the initial stage of emerging infectious diseases such as the COVID-19 epidemic [10].

The case fatality rate (CFR) of COVID-19 cases refers to the ratio of deaths occurring from COVID-19 to the total number of COVID-19 patients within a certain period [11,12]. CFR generally reflects the severity of disease and the comprehensive level of medical diagnosis and treatment capabilities for a certain country [13]. The average of country-/territory-specific COVID-19 CFR is about 2–3% worldwide, which is statistically significantly associated with population size, especially in middle-income and high-income countries, indicating healthcare strain and/or lower treatment efficiency in the countries with large populations, secondary to higher transmission risk, and generally poorer health [14]. A global meta-analysis showed the COVID-19 CFR gradually reached about 4.0~8.0% based on pooled estimates in high-income countries, followed by up to 4.5% In middle-income and between 1.5~3.0% in low-income countries [15]. CFR depends on not only some known factors including age or sex but also recent need for medical care [16]. CFR was related to a variety of country-specific factors, including time to implement social distancing measures after the 100th case, hospital beds per 1000 individuals, percentage of population aged 70+ years, CT scanners per 1 million individuals, and smoking prevalence with high population density. More importantly, the COVID-19 CFR was found to be neither a fixed nor static value, but rather, a dynamic estimate that changes with time, population, socioeconomic factors, and the mitigatory efforts of individual countries [15]. Prediction methods for minimal CFR, which marked national IRC against major infectious diseases such as SARS, Ebola, and COVID-19, to our knowledge, have never been reported.

We hypothesized that the spontaneous minimal CFR values for some nations or regions in the initial stage of infectious disease emergence could be predicted based on multi-disciplinary factors covering social–economical–demographical–environmental characteristics and health-related indicators on national levels, under the scope of one-health. To explore the possible evaluation of national IRC against COVID-19, we tried applying daily CFR data available from 214 nations and regions worldwide to present a deep insight into this problem (Appendix A). We found that the IRC for most nations and regions could be evaluated partially with features of turning points within the daily CFR curve line of COVID-19 patients during the first 40 days of COVID-19 epidemic in each nation (Figure 1). The minimal CFR, a pivotal parameter illustrating the turning points within the initial stage, could be predicted with two established models based on national indicators. The early evaluation of IRC based on minimal CFR prediction in the initial stage might provide essential support to the establishment of suitable national response strategies against other later epidemics for most nations and regions.

## 2. Materials and Methods

### 2.1. Data Source of Actual Daily CFR for All Nations

The available daily numbers of laboratory-confirmed cases and death cases of each country were collected from the WHO website with data sources from national health departments or institutes worldwide. Only the daily CFR data at the beginning of the 2020 COVID-19 pandemic could be utilized due to the focused modelling of minimal CFR in the initial stage. These concrete data from 214 nations and regions—issued every day at 12 o’clock in the Beijing time zone, from the date of the first confirmed case reported to 30 April 2020—were collected. A total of 3,267,184 confirmed cases and 229,971 death cases from 214 nations and regions were enrolled for further analysis. The daily data were accumulated with the longest, 113 days, in China and the shortest, 2 days, in Tajikistan and Comoros. All the cases and related deaths in all nations or regions were caused by the same original strain of SARS-CoV-2 in this study.

### 2.2. Establishment Method of National Actual Daily CFR Curve

The date of the first death case occurring for each nation was marked as Day (T0). The daily CFRs were documented continuously with the cumulative number of daily deaths divided by the cumulative number of confirmed cases per day since Day (0). Considering that daily CFRs show the severity of coronavirus and existing medical treatment capacity day by day, six-degree polynomial regression was used to model CFR curves on country level based on multi-stage waves theory [17]. To avoid possible inconsistency in reporting time and artificial variation of the daily data worldwide, three-day moving arithmetic average CFRs (3DMA-CFR) were used to represent daily CFRs for actual modelling of national CFR curves of COVID-19 epidemics based on the following inclusion criteria: (a) nations and regions had lag days since Day (0) longer than 28 days, (b) nations and regions showed daily reporting data breaks less than three days, (c) nation had more than 20 cumulative deaths from COVID-19 cases, (d) coefficient of determination of national fitting CFR curves was higher than 0.6000, and (e) acceptable data quality of daily confirmed cases and daily deaths. Therefore, all the fitting 3DMA-CFR curves for 92 nations and regions were obtained for next analysis.

### 2.3. Characterization Method of Turning Points in Actual CFR Curves

We explored the lowest actual CFR values in the initial stage from the actual CFR curves for all nations available in early 2020. After extensive comparison of actual 3DMA-CFR curves of 92 nations and regions, we found that all the curves could be classified into curves with or without turning points during the initial stage [18]. There was a falling zone marked with turning points (high point and low point) within the curve for each of 71 nations and regions. Meanwhile, the 3DMA-CFR curves without turning points rose continuously for the other 21 nations and regions.

To quantitatively characterize the falling zone within the fitting curve based on the date of the first death due to COVID-19 occurring for each nation (T0), we first tried to measure the duration in days and CFR decline between two turning points. The characteristic parameters included the day value at the first high point (T1) and the day value at the second low point (T2) after T0, the estimated CFR value at the first high point (CFR1), and the estimated CFR value at the second low point (CFR2). Then, we deduced several characteristic parameters of the falling zone with the day duration and the CFR distance, named ΔT and ΔCFR, respectively (Figure 2). The rectangle square of the falling zone (SFZ) was estimated with the product of ΔT and ΔCFR, which represented the overall spontaneous response capacity against COVID-19 deaths on a national level.

According to the 3DMA-CFR curves during the 40 days after the first death of COVID-19 cases in 71 nations and regions, the representative curves were categorized based on the turning points in every CFR curve. Then, K-M cluster analysis was conducted to explore the kinds of national CFR curves based on the special parameters—including T1, T2, ΔT, CFR1, CFR2, ΔCFR, and SFZ—of turning points within the curves.

### 2.4. Candidate National Indicators Collected for Minimal CFR Prediction

We tried to collect some national information or indicators that might be related to the CFR from the latest data released by United Nations, World Bank, and WHO. These national indicators covered multidisciplinary characteristics such as population, economy, health, and administration, which were assumed to be the potential influencing factors related to the characteristic parameters of national IRC during the initial stages of epidemics. A total of 12 candidate national indicators, including mid-year population estimates (POPU), percentage of population aged 60+ years (A60), percentage of population aged 65+ years (A65), deaths per 1000 people (D1K), births per 1000 people (B1K), GDP per capita (GDP), current health expenditure (HGDP), domestic general government health expenditure (DGDP), physicians per 1000 people (P1K), nurses per 1000 people (N1K), BMI mean of population aged 18+ years old (BMI), and household size (HHS), were explored in this study.

To explore the possible influencing national indicators related to falling zones within 3DMA-CFR curves of COVID-19 epidemics in 65 nations and regions, 7 dependent variables (T1, T2, ΔT, CFR1, CFR2, ΔCFR, SFZ) and 12 independent national indicators related to the economy, population, health administration, financial budget, et al., were explored with Spearman correlation analysis. Only CFR2 was found to have significant correlations with some independent variables.

### 2.5. Prediction Method of Minimal CFR on Nation Levels

Among the 71 nations and regions, 6 nations including Algeria, Indonesia, Philip-pines, San Marino, Sudan, and the United Kingdom had CFRs higher than 7 percent, which is rare for epidemics, and were excluded. Therefore, the other 65 nations and regions were included for subsequent modelling analysis. To find the factors contributing to the pivotal parameter (CFR2) of falling zone within daily CFR curves, multiple linear regressions and multi-order curve regressions were applied separately.

To avoid multicollinearity during the subsequent multiple linear regression analysis, all nine independent variables were first transferred into several principal components. Four principal components were extracted successfully after principal component analysis, and the cumulative proportion was over 85%. Only one sensitive principal component was selected to enter the following regression model (M1).

To explore a better-fitting regression model, multivariate curve regressions of the second-order, third-order, and fourth-order were applied step by step. Using the above similar nine independent variables of 65 nations and regions, another multi-order regression model (M2) was established with acceptable fitting performance.

### 2.6. Evaluation of Prediction Models for National Minimal CFR in the Initial Stage

To check the prediction efficiency of the two established models (M1 and M2), the national indicators of 65 nations and regions were used to calculate minimal CFR before April 30 for internal evaluation. With the difference of national minimal CFR calculated from each model and national CFR2 deduced from the national 3DMA-CFR curves, the parameters such as root mean squared error (RMSE), mean absolute error (MAE), relative absolute error (RAE), root relative squared error (RRSE), and coefficient of determination (R^2^) were applied to evaluate the prediction performance of the two models.

To check prediction efficiency of these two models more comprehensively, the above similar nine independent variables of 73 other nations and regions were applied for external evaluation based on the following COVID-19 data after 30 April 2020 on the WHO website. Among them, 26 nations and regions failed to build the 3DMA-CFR curves because they were beyond the inclusion criteria (Section 2.2), 20 nations and regions were abandoned for further analysis because they had COVID-19 deaths fewer than 20 cumulative cases, and 10 nations and regions were abandoned because of the poor fitting performance with R^2^ lower than 0.60. Thus, only 17 nations and regions with turning points within 3DMA-CFR curves, including the Central African Republic, Chad, French Guiana, Guinea, Haiti, Kuwait, Lesotho, Liberia, Madagascar, Namibia, Nepal, Sierra Leone, Slovakia, Somalia, South Sudan, Tajikistan, and Yemen, were selected for further external evaluation. The CFR2 and national indicators used for external evaluation of prediction efficiency were similar to those for internal evaluation.

### 2.7. Quality Assessment for Global CFR Data Source

Python software was used to download data from WHO website. The data quality might be influenced by a few factors such as the difference of reporting regulations in different nations and regions. In order to assure the quality of data analyzed, we evaluated the difference of daily death cases from the WHO website and DXY website, an authoritative medical internet platform in China, for 20 consecutive days regarding to COVID-19 outbreaks in China, Italy, Spain, and Germany. As early as January 21, 2020, DXY first launched an epidemic map to collect global data from WHO and government agencies of various nations and regions [19]. The difference of all daily data pairs (WHO versus DXY) regarding to the confirmed and death cases in the four nations and regions above were all less than 1.0%, which implied that the data collected had almost no day lag, and therefore were reliable and qualified for further analysis.

### 2.8. Statistical Analysis

ANOVA testing was used to explore the parameter difference of falling zones with SPSS 17.0. Spearman correlation testing was applied to screen 12 independent variables related to 8 dependent variables. The independent variables were screened by principal component analysis to eliminate collinearity between the independent variables, and the extracted principal component were used for principal component regression analysis. By transforming standardized independent and dependent variables, the relationship was expressed as an arithmetical model to predict the minimal CFRs in the spontaneous initial stages of COVID-19 epidemics.

## 3. Results

### 3.1. Actual Daily 3DMA-CFR Curves of COVID-19 Epidemics in 214 Nations

The data of daily confirmed cases and daily deaths of initial COVID-19 epidemics in 214 nations and regions, including 61 in Europe, 54 in the North America and South America, 48 in Africa, 22 in the eastern Mediterranean, 19 in the western Pacific, and 10 in Southeast Asia, were collected continuously from the WHO (Appendix A). The respective 3DMA-CFR values were used to fit the daily CFR regression curves successfully based on six-degree polynomial regression in 92 nations and regions (Appendix A) with 122 nations or regions excluded by the five inclusion criteria. All correlation coefficients of 90 nations and regions were higher than 0.8000. The coefficient of determination (R^2^) for Afghanistan and the United Arab Emirates were 0.6916 and 0.7854. The other 122 nations and regions were abandoned for further analysis because they did not match the including criteria.

### 3.2. Categories of Actual 3DMA-CFR Curves of COVID-19 Epidemics in 92 Nations

Four kinds of 3DMA-CFR curves were found in 92 nations and regions based on the containing turning points (Figure 3): (a) The curves of 21 nations and regions, including Brazil and Spain, showed a rising 3DMA-CFR without turning points. (b) The curves of some nations and regions, including Turkey and the USA, showed a rising 3DMA-CFR and similar falling zones marked with the first high turning points and the second low turning points. (c) The curves of some nations and regions, including Denmark and Columbia, showed the first falling and then a rising 3DMA-CFR based on the low turning points marked with one or more waves. (d) The curves of some nations and regions, including Argentina and Iran, showed the first falling and then the second massively rising 3DMA-CFR based on the low turning points marked with the minimal CFRs. Among the 92 nations and regions, there were 71 nations and regions with similar falling zones marked with the high turning points and low turning points, which were selected for next exploration.

### 3.3. Characteristic Parameters of Actual 3DMA-CFR Curves for COVID-19 Epidemics

The characteristic parameters of falling zones were estimated with T1, T2, CFR1, CFR2, ΔT, ΔCFR, and SFZ according to the turning points within 3DMA-CFR curves for 71 nations and regions (Table 1). As for the day duration, the ranges of T2 and ΔT were 0.2586~53.1038 days and 0.2586~40.2292 days. Iraq and Congo showed the maximal and minimal ΔTs. As for the CFR decline, the ranges of CFR2 and ΔCFR were 0.0682~32.5804 percent and 0.0005~82.4768 percent. New Zealand and the Philippines showed the lowest and highest CFR2. Sudan, Iran, and Ukraine ranked as the top three nations and regions with the largest SFZs.

### 3.4. Classification of Falling Zones within Actual Daily CFRs Curves

Cluster analysis based on seven parameters including T1, T2, CFR1, CFR2, ΔT, ΔCFR, and SFZ showed that all the falling zones within 3DMA-CFR curves of COVID-19 epidemics for 71 nations and regions can be divided into three categories. Cluster 1 marked with lowest CFR2 included 51 nations and regions, such as Austria, Canada, and Denmark. Cluster 2 marked with moderate CFR2 included 16 nations and regions, such as Australia, China, and USA. Cluster 3 marked with highest CFR2 included four nations and regions, such as Iran, Sudan, and Ukraine (Figure 4). Kruskal–Wallis H test analysis revealed that there was significant difference for all seven parameters among three groups of different clusters (*p* < 0.05) (Table 2).

### 3.5. Contributing Factors to the Lowest CFR2 within Actual 3DMA-CFR Curves

Among the above 71 nations and regions, 6 nations and regions including Algeria, Indonesia, Philippines, San Marino, Sudan and the United Kingdom were excluded for further modelling because of the rare high CFR2 above 7 percent. Among seven dependent variables (T1, T2, ΔT, CFR1, CFR2, ΔCFR, SFZ), Spearman correlation analysis showed only CFR2 had significant correlations with nine independent variables including Log(POPU), Log(GDP), A65, B1K, DGDP, P1K, N1K, BMI, and HHS in 65 nations and regions.

The PCA analysis of all the nine independent variables showed that only the first principal components entered the following regression model (*p* = 0.0108). The multiple linear regression model equation (M1), after the transformation of standardized dependent and independent variables, showed that national indicators including Log (POPU), B1K, and HHS showed significant positive contributions, and national indicators including Log (GDP), A65, DGDP, P1K, N1K, and BMI (21.8~29.5) showed negative contributions to the minimal CFR in the initial COVID-19 epidemics for most nations and regions.

Based on the above same nine independent variables of 65 nations and regions, the following model from fourth-order curve regression (M2) was established with better fitting performance (R^2^ = 0.7106). National indicators including Log(POPU), Log(GDP), A65, DGDP, B1K, P1K, N1K, HHS, and BMI showed complex negative or positive contributions to the minimal CFR of initial COVID-19 epidemics for most nations and regions.

Model M1: minimal CFR = 5.80353205 + 0.13769617 × Log (POPU) − 0.33097963 × Log (GDP) − 0.02860491 × A65 + 0.02046821 × B1K − 0.02103376 × DGDP − 0.11903786 × P1K − 0.04488684 × N1K − 0.07041279 × BMI + 0.13482426 × HHS

Model M2: minimal CFR = 9119 − 5.419 × Log(POPU) + 1190 × Log(GDP) − 0.4005 × A65 − 0.3356 × B1K + 1.647 × DGDP − 0.284 × P1K − 0.1981 × N1K − 1601 × BMI + 4.969 × HHS + 8.974 × Log(POPU)^2^ − 456.9 × Log(GDP)^2^ + 0.05642 × A65^2^ + 0.02057 × B1K^2^ − 0.1449 × DGDP^2^ − 0.03604 × P1K^2^ + 0.01743 × N1K^2^ + 93.2 × BMI^2^ − 1.363 × HHS^2^ − 5.01 × Log(POPU)^3^ + 77.18 × Log(GDP)^3^ − 0.003278 × A65^3^ − 0.0006485 × B1K^3^ + 0.005122 × DGDP^3^ + 0.01372 × P1K^3^ − 0.00166 × N1K^3^ − 2.402 × BMI^3^ + 0.09594 × HHS^3^ + 0.8692 × Log(POPU)^4^ − 4.845 × Log(GDP)^4^ + 0.00006483 × A65^4^ + 0.000009345 × B1K^4^ − 0.00006005 × DGDP^4^ − 0.0005694 × P1K^4^ + 0.00007282 × N1K^4^ + 0.02313 × BMI^4^ + 0.002492 × HHS^4^

Note: POPU: mid-year population estimates; A65: percentage of population aged 65+ years; B1K: births per 1000 people; GDP: GDP per capita; DGDP: domestic general government health expenditure; P1K: physicians per 1000 people; N1K: nurses per 1000 people; BMI: BMI mean of population aged 18+ years old; HHS: average household size.

### 3.6. Prediction Efficiency of Minimal CFR Models from Internal Evaluation

Using the national indicator data of 65 nations and regions to predict minimal CFR, the results from internal evaluation showed that the minimal CFRs calculated with the above two regression models showed good concordance for nearly 60 nations and regions.

With model M1, the minimal CFR was predicted successfully for 57 nations and regions based on the nine national indicators (R^2^ = 0.5074) (Figure 5a). RMSE was 1.2978% and MAE was 1.0788%. RAE was 1.1949 and RRSE was 1.1287. There was poor prediction performance for the other eight nations and regions, including Bolivia, Cameroon, Kuwait, Nigeria, Pakistan, Puerto Rico, Saudi Arabia, and Thailand. The possible reasons were supposed to be related to low R^2^ values of 3DMA-CFR curves based on the raw data of poor quality from some nations and regions such as Cameroon (R^2^ = 0.8193), Puerto Rico (R^2^ = 0.8585), and Bolivia (R^2^ = 0.8899).

With model M2, CFR2 was predicted successfully for 59 nations and regions based on the nine national indicators (R^2^ = 0.8008) (Figure 5b). RMSE was 0.6562% and MAE was 0.5107%. RAE was 0.5017 and RRSE was 0.5092. There was poor prediction performance for six nations and regions including Ecuador, France, Lebanon, Lithuania, Russia, and Turkey. All the R^2^ values of 3DMA-CFR curves for those six nations and regions showed relative lower. The R^2^ values of Turkey and Ecuador were 0.8445 and 0.8752, respectively.

### 3.7. Prediction Efficiency of Minimal CFR Models from External Evaluation

For the 17 nations and regions at external evaluation, each actual national CFR2 value was deduced from the national 3DMA-CFR curve line (Appendix A). The national indicator data were absent for four nations and regions, including French Guiana, Kuwait, South Sudan, and Yemen, and invalid to be calculated for prediction with these two models. For some of the other 13 nations and regions, the results from the external evaluation showed that the CFR2 could be predicted with the above two models with good concordance.

With model M1, the minimal CFR was predicted successfully for nine nations and regions, including Chad, Haiti, Liberia, Namibia, Nepal, Sierra Leone, Slovakia, Somalia, and Tajikistan, based on the similar nine national indicators (R^2^ = 0.6826) (Figure 6a). Without Namibia, minimal CFR was predicted more accurately for eight nations and regions (R^2^ = 0.8789) (Figure 6b). RMSE was 2.3576% and MAE was 1.9542%. RAE was 0.6216 and RRSE was 0.7008. There was poor prediction performance for four nations and regions in Africa, including the Central African Republic, Guinea, Lesotho, and Madagascar.

With model M2, the minimal CFR was predicted successfully for only five nations and regions, including Guinea, Lesotho, Madagascar, Namibia, and Chad, based on the nine national indicators (R^2^ = 0.9146). RMSE was 1.2281% and MAE was 1.0300%. RAE was 0.3216 and RRSE was 0.3674. There was poor prediction performance for the other eight nations and regions, including the Central African Republic, Haiti, Liberia, Nepal, Slovakia, Sierra Leone, Somalia, and Tajikistan.

## 4. Discussion

Here we provided a pilot study of turning points contained in the fitted daily CFR curves. The model to predict the minimal CFR value for initial COVID-19 epidemics was first reported based on multidisciplinary characteristics or indicators at the national level, which theoretically provides an important tool to evaluate the integrated responding capacity against emerging infectious disease in advance for any nation.

It is very difficult to establish suitable national response strategies against emerging infectious disease for any nation, especially in the initial stages. There was occasionally reluctance or over-confidence of the IRC including existing medical treatment systems [20]. The outbreak of the COVID-19 pandemic emphasized the need to predict the minimal CFR in the initial stage of any future epidemics for all nations [21,22]. We noticed that the information of national IRC might be contained within the daily CFR curves, but how to predict minimal CFR value for initial novel epidemics for each nation in the past and future was much more challenging. We confirmed this challenging hypothesis about the spontaneous minimal CFRs in the initial stage for individual nation could be predicted.

This study was exceeding difficult to carried out because of the following four considerations. (a) The modelling needed enough original data at the national level and only the global condition emerging from the 2020 COVID-19 pandemic was available for such a study. (b) The data on emerging epidemics in all nations or regions should be from unique virus pathogens. The original strain of COVID-19 in early 2020 was the same SARS-CoV-2 virus globally. (c) The 2020 COVID-19 epidemic in all nations provided reliable data based on individual PCR measurement for each infected case and related death cases in recent years. (d) This study focused on the minimal CFR prediction for nations or regions in the initial stage of a novel epidemic, which implied the daily CFR data of 2020 COVID-19 epidemic were collected under conditions without solid social intervention, quarantine [20], and isolation measures, especially without extensive vaccination [23,24].

After extensive observation and artificial comparison of fitted CFR curves, we found a similar tendency in daily CFR curves for many nations and regions and tried to explore the inner key information of turning points within the daily CFR curves for COVID-19 epidemics, especially in the initial stages. As for the possible reasons of these two points, we presented a somewhat reasonable but subjective explanation here according to wave theory in this pilot study [17]. There were several stages if public health interventions were not implemented correctly and in a timely manner [25]. At the initial stage, the daily CFRs of COVID-19 cases arose generally due to the absence of social response measures in almost all countries. At the second stage—after a few days, whether in developed or developing countries—the national CFRs would be lower, usually from a high point, because of spontaneous or intensive response from existing medical systems. At the third stage, sooner or later, the CFRs would arise at a low point because of the possible exhausting of national IRC. Given a lack of specific public health intervention, this stage—with a possible falling zone—was very important for the tendency of the COVID-19 epidemic for each nation, which implies how strong the national basic IRC was in defending against the attack of COVID-19 epidemics in the initial stage.

Four kinds of daily CFR curves were found in this study. Some curves, marked by a lack of any turning points and absence of falling zone, were defined with negative IRC in nations and regions such as Spain, which were characterized by insufficient medical systems [26]. The curves showed the CFRs increased continuously without a turning point in the initial 40 days.

Among the other nations and regions with positive IRCs, the fitting curves showed a different style in each nation. The curves of category B were defined by moderate national IRC in nations and regions such as the United States, which were characterized by general medical measures marked with general day duration and/or CFR decline and, especially, a middle rectangle in the falling zone. The curves showed several waves, and the final CFR was maintained or slightly lower than the highest CFR. The curves of category C were defined by weak IRC in nations and regions such as Denmark, which was characterized by limited medical measures marked with short CFR decline, and especially a minor rectangle in the falling zone. The CFR reversed and continued to rise after a transient drop from the first high point. The curves of category D, with strong national IRC against COVID-19 epidemics, in nations and regions such as Cuba characterized with effective medical measures were marked by long day duration and/or CFR decline and, especially, a large rectangle in the falling zone. For national curves with type A, the IRC of these countries showed almost no contributions against COVID-19 epidemic. For national curves with type D, the IRC of these countries played important roles to lag CFRs of COVID-19 cases with long day duration. For most nations and regions, the IRC were observed to be exhausted at this pivotal point which was marked by the minimal CFR, implying a worsening or even destruction of the national IRC. Therefore, we deduced that the minimal CFR within the 3DMA-CFR curve was closely related to the national IRC against COVID-19 epidemics.

We demonstrated the possibility of minimal CFR prediction with models and the minimal CFR was confirmed to be related to multiple factors regarding national health policy, social-economic policy, age level, and demographical indicators, based on the modelling in this study. The basic reproduction number did not vary significantly in different regions of the world for epidemic transmission of the unique virus variant or pathogen [27]. A Dynamic Bayesian Network (DBN) to predict community-level CFR of COVID-19 infection in USA was explored based on some measures of social and environmental vulnerability, including temperature and humidity [28]. The daily average temperature, humidity, and wind speed negatively affected the daily new death cases in early 2020 in Brazil [29]. Moreover, the CFR estimation of COVID-19 was 2.67 based on a meta-analysis of 45 studies, and the average CFRs in different regions of the world varied significantly, from 2.49 in Asia to 3.40 in North America. Higher CFR values were found in countries with insufficient PCR testing and greater median population age, which implies the potential influence of multiple national factors from social–economical–environmental–medical aspects [27].

The other six parameters including T1, T2, ΔT, CFR1, ΔCFR, and SFZ failed to produce correlations and establish models. Considering the possible poor reporting efficiency for early death cases (T0), parameters including T1, T2, ΔT, and even the SFZ might have varied much and therefore become less credible. Compared with CFR2, CFR1 might be less stable for individual nations because of the smaller denominator.

In this study, the data used were explored from 214 nations and regions—as many as possible—and 71 nations and regions were applied to establish models with serial screening. Models M1 and M2 both showed good fitting performance for the internal evaluation of 65 nations and regions based on the daily cases before 30 April. As to the external evaluation, only the model M1 showed good fitting performance for nine nations and regions based on the daily cases after 30 April. As to the poor performance of model M2 for the external evaluation, the absence of reasonable principal for modelling might be one of the possible reasons.

Based on the innovative models in this study, we would present some valuable advice for minimal CFR estimation in the initial stages on the national level. For a brand-new epidemic with a unique virus strain pathogen, in the future, the prediction model M1 revealed that the minimal CFR would be higher for nations with the three leading risk factors including larger population, higher average household size in all families, and larger births number per 1000 people. Some other national indicators, including higher GDP, more physicians and nurses per 1000 people and even higher body mass index, would make minimal CFR estimation much lower interestingly.

There were several limitations in this modelling study. There was obvious variation of diagnosis ability among the 214 nations and regions. The accuracy of daily data collected varied, also, in different nations or regions due to possible underreporting. Moreover, the time delay of daily data indeed existed and was inevitable in not just one nation; this was why we preferred a three-day moving average CFR for conducting analysis [30]. It is necessary to confirm the first pilot predictions of minimal CFR with future repetitive studies, including the trying of non-polynomial and symbolic regression, because the recorded cases were under-sampled in some nations.

## 5. Conclusions

In the initial stage of emerging infectious epidemics such as COVID-19, the 3DMA-CFR curve can revert to rising at the low turning point marked with minimal CFR because the national IRC has become exhausted quickly in many nations and regions. This study presented a novel modelling method based on multidisciplinary factors on nation levels to predict the minimal CFR, which sheds light on the quantitative evidence required to establish suitable response strategies worldwide, especially for nations without death cases in the initial dozens of days.

## Figures and Tables

**Figure 1 ijerph-20-00594-f001:**
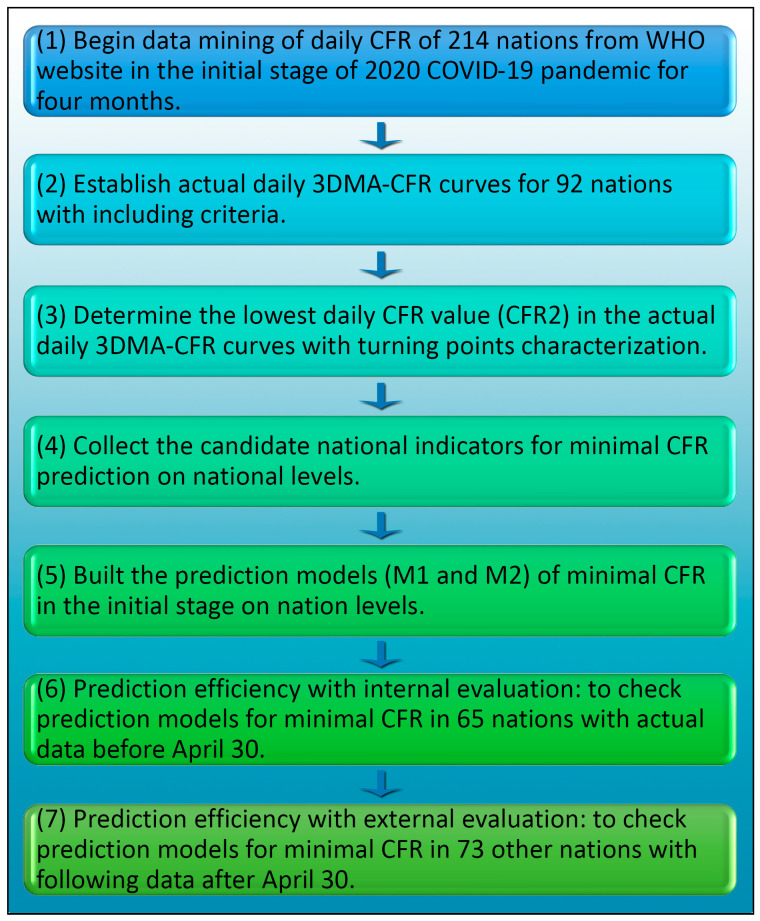
Illustration for modelling and prediction of national minimal CFR value in the initial stage of epidemic with national indicators.

**Figure 2 ijerph-20-00594-f002:**
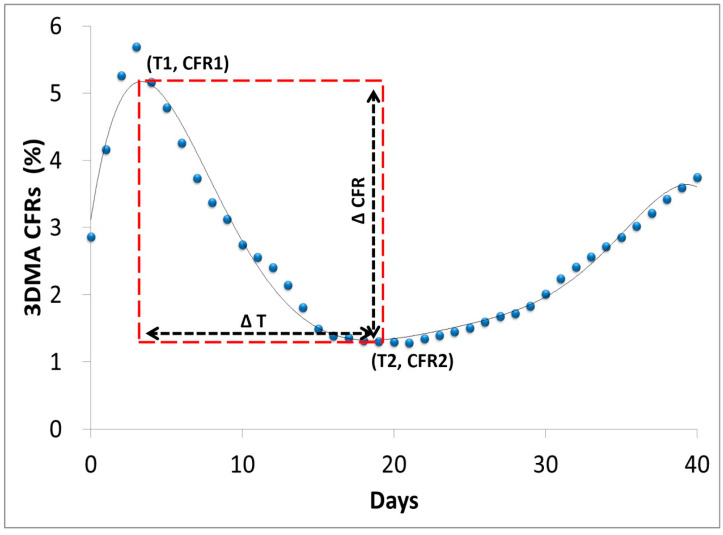
Quantitative characterization of falling zone within common national fitting CFR curve of COVID-19 epidemics.

**Figure 3 ijerph-20-00594-f003:**
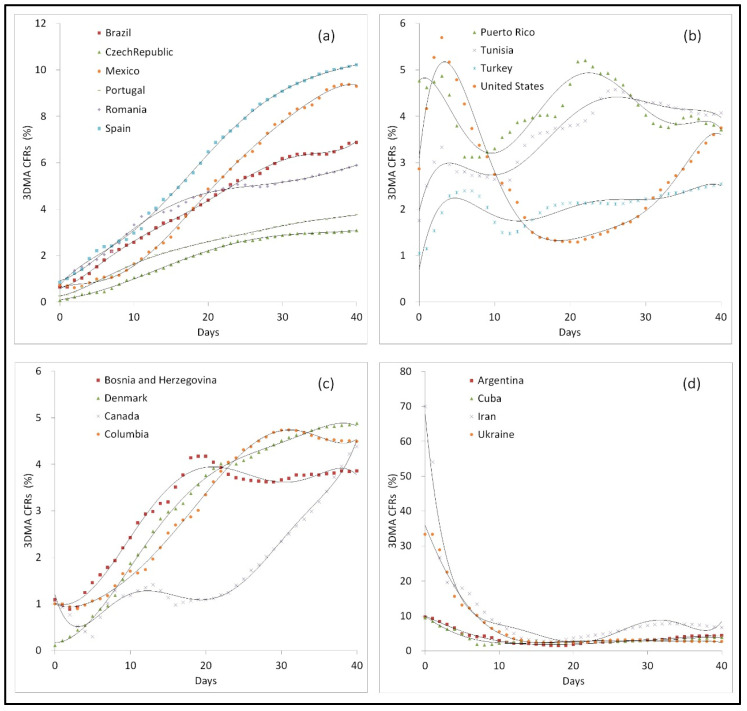
Representative national 3DMA-CFR curves for COVID-19 epidemics. (**a**) The arising curves without turning points. (**b**) The curves containing similar falling zones marked with the first high turning points and the second low turning points. (**c**) The curves showing the first falling and then the second rising based on the low turning points. (**d**) The curves showed the first falling and then the second massively rising based on the low turning points.

**Figure 4 ijerph-20-00594-f004:**
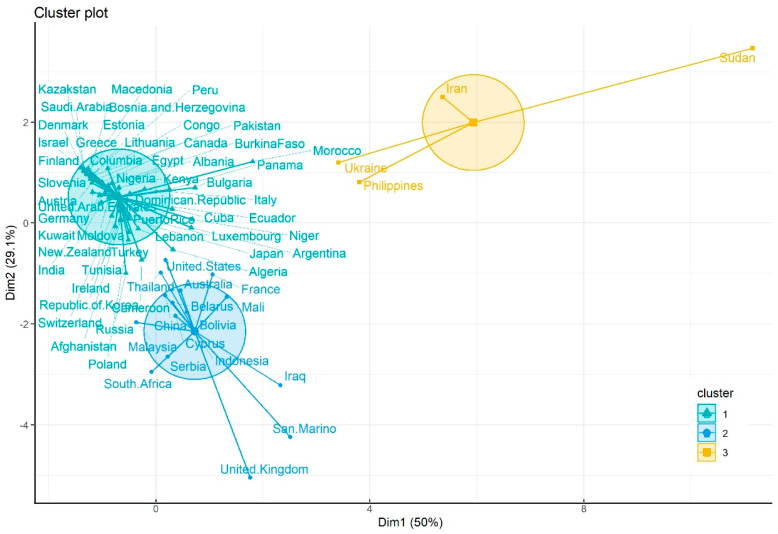
Three clusters based on seven parameters of turning points within 3DMA-CFR curves of initial COVID-19 epidemics for 71 nations.

**Figure 5 ijerph-20-00594-f005:**
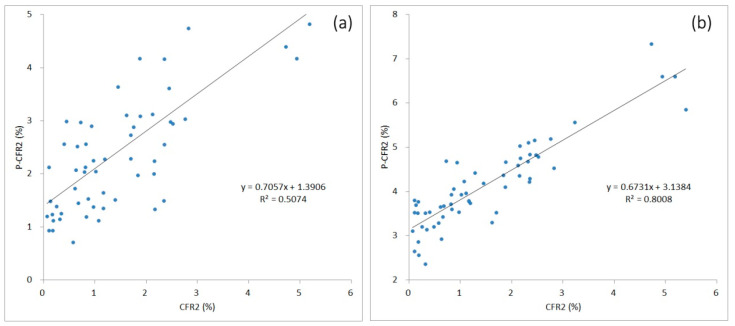
Prediction performances of CFR2 with two models for initial COVID-19 epidemics for internal evaluation. CFR2 values: calculated from existing national CFR curves. Predicted CFR2 (p-CFR2) values: estimated from model M1 (**a**) and model M2 (**b**).

**Figure 6 ijerph-20-00594-f006:**
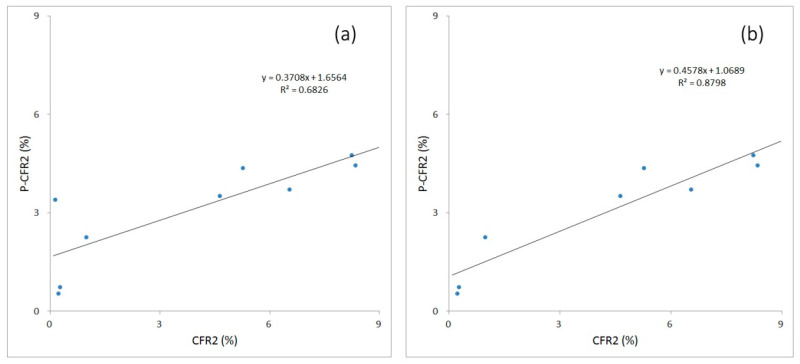
Prediction performance of CFR2 for initial COVID-19 epidemics with established model M1 for nine nations (**a**) or eight nations (**b**) in external evaluation. CFR2 values: calculated from the existing national CFR curves. p-CFR2 values: predicted from model M1.

**Table 1 ijerph-20-00594-t001:** Characteristic parameters of 3DMA-CFR curves in initial COVID-19 epidemics for 71 nations and regions.

No.	Nations	R^2^	T1 (Day)	T2 (Day)	CFR1 (%)	CFR2 (%)	ΔT (Day)	ΔCFR (%)	SFZ	Clusters
1	Morocco	0.974	0.000	10.579	27.765	2.440	10.579	25.325	267.900	1
2	Bulgaria	0.947	0.000	11.214	15.752	1.166	11.214	14.586	163.567	1
3	Argentina	0.986	0.000	17.232	10.110	1.700	17.232	8.410	144.923	1
4	Niger	0.929	2.473	15.341	12.859	2.317	12.869	10.541	135.650	1
5	Panama	0.988	0.000	9.855	11.889	0.822	9.855	11.067	109.066	1
6	Cuba	0.940	0.000	12.632	9.997	2.170	12.632	7.827	98.872	1
7	Italy	0.986	0.000	7.345	8.500	2.342	7.345	6.158	45.230	1
8	Ecuador	0.875	0.000	8.726	4.124	0.447	8.726	3.677	32.085	1
9	Algeria	0.898	6.369	16.208	9.410	7.024	9.839	2.386	23.471	1
10	Lebanon	0.902	1.931	13.313	3.646	1.697	11.382	1.949	22.185	1
11	Luxembourg	0.948	0.000	11.097	2.780	0.830	11.097	1.950	21.637	1
12	DO	0.899	0.000	6.952	5.561	2.517	6.952	3.044	21.161	1
13	Cameroon	0.819	7.149	18.097	2.974	1.104	10.948	1.870	20.467	1
14	Puerto Rico	0.859	0.557	9.611	4.831	3.224	9.054	1.607	14.551	1
15	Japan	0.809	0.000	11.549	2.359	1.168	11.549	1.191	13.758	1
16	Ireland	0.995	0.000	9.976	1.651	0.319	9.976	1.332	13.284	1
17	Kenya	0.911	0.000	5.216	4.470	2.355	5.216	2.116	11.036	1
18	Poland	0.994	2.507	13.435	2.008	1.013	10.928	0.995	10.877	1
19	UAE	0.785	0.000	5.779	1.424	0.656	5.779	0.768	4.440	1
20	Moldova	0.972	0.000	6.823	1.467	0.817	6.823	0.650	4.432	1
21	Turkey	0.845	5.052	13.930	2.215	1.755	8.877	0.460	4.083	1
22	Burkina Faso	0.823	0.000	3.861	3.674	2.819	3.861	0.854	3.299	1
23	Nigeria	0.964	0.000	4.303	2.001	1.287	4.303	0.715	3.075	1
24	Egypt	0.989	0.000	3.562	2.093	1.446	3.562	0.647	2.304	1
25	Albania	0.969	0.000	4.110	2.834	2.351	4.110	0.484	1.988	1
26	Afghanistan	0.692	3.574	10.147	2.609	2.322	6.573	0.288	1.890	1
27	Switzerland	0.996	1.923	9.824	0.741	0.575	7.901	0.166	1.310	1
28	Tunisia	0.945	4.158	9.656	2.970	2.760	5.498	0.211	1.159	1
29	Canada	0.994	0.000	3.066	0.982	0.673	3.066	0.308	0.945	1
30	New Zealand	0.994	1.397	7.220	0.209	0.068	5.824	0.141	0.819	1
31	Kuwait	0.998	0.000	6.064	0.197	0.102	6.064	0.095	0.573	1
32	ROK	0.990	3.069	11.577	0.674	0.629	8.508	0.045	0.383	1
33	Slovenia	0.998	0.000	2.693	0.382	0.256	2.693	0.127	0.342	1
34	Pakistan	0.989	0.000	2.215	0.628	0.486	2.215	0.142	0.314	1
35	Russia	0.969	11.602	18.631	0.831	0.793	7.029	0.038	0.268	1
36	Finland	0.994	0.000	1.849	0.248	0.105	1.849	0.143	0.264	1
37	Estonia	0.994	0.000	1.577	0.288	0.134	1.577	0.154	0.243	1
38	Peru	0.859	0.000	1.913	0.843	0.722	1.913	0.121	0.231	1
39	Lithuania	0.967	0.000	1.989	1.079	0.969	1.989	0.111	0.220	1
40	Austria	0.999	0.000	3.463	0.218	0.169	3.463	0.049	0.169	1
41	India	0.937	0.000	4.968	1.902	1.877	4.968	0.026	0.129	1
42	Colombia	0.998	0.000	1.899	1.002	0.938	1.899	0.064	0.122	1
43	BiH	0.986	0.000	1.452	1.052	0.975	1.452	0.077	0.111	1
44	Saudi Arabia	0.979	0.000	1.070	0.223	0.179	1.070	0.044	0.047	1
45	Denmark	0.999	0.000	0.805	0.215	0.189	0.805	0.026	0.021	1
46	Greece	0.994	0.000	0.695	0.887	0.874	0.695	0.013	0.009	1
47	Germany	1.000	2.176	4.718	0.179	0.177	2.542	0.002	0.005	1
48	Kazakhstan	0.970	0.000	0.636	0.402	0.396	0.636	0.007	0.004	1
49	Macedonia	0.984	0.000	0.510	1.196	1.190	0.510	0.006	0.003	1
50	Congo	0.944	0.000	0.259	4.724	4.716	0.259	0.009	0.002	1
51	Israel	0.996	0.000	0.276	0.107	0.106	0.276	0.001	0.000	1
52	Iraq	0.869	6.687	46.916	9.323	4.930	40.229	4.393	176.725	2
53	France	0.974	3.357	26.049	9.081	1.396	22.692	7.686	174.399	2
54	San Marino	0.865	20.285	53.104	11.713	7.126	32.819	4.588	150.557	2
55	Mali	0.957	4.693	28.833	10.482	5.179	24.140	5.303	128.019	2
56	Australia	0.980	3.259	26.814	3.460	0.349	23.554	3.111	73.282	2
57	USA	0.970	3.411	20.453	4.755	1.070	17.042	3.685	62.793	2
58	Thailand	0.984	2.550	22.637	2.388	0.314	20.087	2.074	41.668	2
59	Bolivia	0.890	13.003	27.152	7.384	5.394	14.149	1.990	28.157	2
60	Belarus	0.940	2.892	28.198	1.453	0.610	25.306	0.843	21.330	2
61	Cyprus	0.898	7.991	29.126	2.872	2.148	21.135	0.725	15.313	2
62	China	0.956	6.721	25.215	2.859	2.119	18.494	0.740	13.688	2
63	Serbia	0.902	20.422	33.525	2.351	1.838	13.102	0.512	6.714	2
64	Indonesia	0.863	17.804	28.588	9.008	8.567	10.784	0.441	4.755	2
65	UK	0.993	44.515	50.498	13.674	13.386	5.984	0.288	1.723	2
66	South Africa	0.991	29.803	33.392	1.896	1.883	3.589	0.013	0.046	2
67	Malaysia	0.991	22.212	25.475	1.619	1.617	3.263	0.002	0.006	2
68	Sudan	0.967	0.000	21.198	100.000	17.523	21.198	82.477	1748.377	3
69	Iran	0.928	0.000	13.668	60.117	2.470	13.668	57.647	787.944	3
70	Ukraine	0.988	0.000	16.979	36.206	2.158	16.979	34.048	578.085	3
71	Philippines	0.930	0.000	8.304	46.899	32.580	8.304	14.319	118.908	3

Notes: R^2^: coefficient of determination for CFR curve equation of each nation. T1: the day of first high point within the CFR curve. T2: the day of second low point within the CFR curve. CFR1: the CFR value of first high point within the CFR curve. CFR2: the CFR value of second low point within the CFR curve.

**Table 2 ijerph-20-00594-t002:** Seven parameters of turning points within 3DMA-CFR curves for 71 nations among three clusters (Mean ± SD).

Groups	*n*	T1 (Day)	T2 (Day)	ΔT (Day)	CFR1 (%)	CFR2 (%)	ΔCFR (%)	STC
Cluster 1	51	1.06 ± 2.29	7.06 ± 5.28	6.00 ± 4.16	3.55 ± 5.04	1.34 ± 1.29	2.22 ± 4.61	23.59 ± 52.46
Cluster 2	16	13.10 ± 11.99	31.62 ± 9.84	18.52 ± 10.10	5.89 ± 4.11	3.62 ± 3.62	2.28 ± 2.29	56.20 ± 64.79
Cluster 3	4	0.00	15.04 ± 5.44	15.04 ± 5.44	60.81 ± 27.90	13.68 ± 14.50	47.12 ± 29.48	808.33 ± 686.15
*p*		<0.001	<0.001	<0.001	<0.001	0.002	0.001	<0.001

Note: Kruskal–Wallis H test of non-parametric test was applied for difference comparison among different clusters.

## Data Availability

Not applicable.

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
