# Peer review of "Predictive Model for National Minimal CFR during Spontaneous Initial Outbreak of Emerging Infectious Disease: Lessons from COVID-19 Pandemic in 214 Nations and Regions"

_ijerph, 2022, doi:10.3390/ijerph20010594_

Round 1

Reviewer 1 Report (Previous Reviewer 1)

This manuscript adds two authors but the format is inconsistency. No more mew comments.

Author Response

Point 1: This manuscript adds two authors but the format is inconsistency. No more mew comments.

Response 1: Thank the reviewer very much for the dedication and strict instruction. We are sorry for the inconsistent expression about authors. The information about two added authors were deleted completely in the section of Author Contributions (P16, L512).

Reviewer 2 Report (Previous Reviewer 2)

While the authors made good progress in improving the manuscript, the methodological errors remain. As such, I would have to suggest rejecting this version as well. 

Author Response

Point 1: While the authors made good progress in improving the manuscript, the methodological errors remain. As such, I would have to suggest rejecting this version as well.

Response 1: Thank the reviewer very much for his insightful comments. We feel a little frustrated for the possible methodological errors in the revised manuscript. Although we made every concrete answer and/or revision about 11 concrete comment or concerning after the first-round reviewing, we should say sorry because we couldn’t solve the remaining methodological errors without concrete instruction or comment.

The spontaneous minimal CFR in the initial stage of any new emerging epidemic is critical and essential fundament for establishment of diverse national responding strategy and less is known about the prediction method and influencing factors of it on nation level. With one rare and exceedingly valuable opportunity from earlier COVID-19 pandemic 2020, this study hypothesized, under the scope of one-health, that the spontaneous minimal CFRs for some nations or regions could be predicted in the initial stage of emerging infectious disease based on multidisciplinary factors covering social-economical-demographical-environmental characteristics and health-related indicators on nation levels. Based on the actual minimal CFR values from the results , the study confirmed that national spontaneous minimal CFR could be predicted with models (M1 and M2) successfully for most nations and regions against COVID-19 epidemics, which provided critical method to predict the essential early evidence to evaluate the integrating responding capacity and establish national responding strategies reasonably for other emerging infectious diseases in the future.

We think this hypothesis about the spontaneous minimal CFRs in the initial stage for each nation is challenging to be predicted, not similar with the average CFRs characterization for whole duration of COVID-19 pandemic in some nations. Meanwhile, this study is exceeding difficult to carry out with rare opportunity because of the following four considerations or preconditions. a) The modelling needed enough original data on nation levels and only the global condition from 2020 COVID-19 pandemic was available for such a study. b) The data of emerging epidemic in all nations or regions should be from the unique virus pathogen, and there was all the same original strain of SARS-CoV-2 virus in early 2020 globally. c) The 2020 COVID-19 epidemic in all nations provided the reliable data based on individual PCR measurement for each infected case and the related death case in recent years. d) This study focused the minimal CFR prediction for nations or regions in the initial stage of novel epidemic, which implied the daily CFR data of 2020 COVID-19 epidemic were collected under conditions without solid social intervention, quarantine and isolation measures, especially without extensive vaccination.

We finished such a pilot study with two novel prediction models of minimal CFR established with multiple linear regressions (M1) and multi-order curve regressions (M2) after internal and external evaluation. These two kinds of regression methods are classical, especially in public health modelling field, and reliable to provide credible result in this study with acceptable prediction performance (RMSE, MAE, RAE and RRSE values). It is really a pity that we are not familiar with some other non-polynomial models and therefore we couldn’t provide more results about non-polynomial modelling. Meanwhile, the necessity with such non-polynomial models should be validated at the current stage.

In fact, we tried multiple modellings method to assure acceptable performance. It is really very difficult to establish such a modelling for minimal CFR value in the initial stage. To avoid the multicollinearity, all the nine independent variables were first transferred into several principal components. Four principal components were extracted successfully after principal component analysis, and the cumulative proportion was over 85%. Only one sensitive principal component was selected to enter the following regression model (M1). To explore better fitting regression model, multivariate curve regressions of second-order, third-order and fourth-order were applied step by step. Moreover, based on the above same nine independent variables of 65 nations and regions, another multi-order regression model (M2) was established with acceptable fitting performance. We asked many scholars majored in mathematic for modelling advise and we reported the best modelling results in the current manuscript.

This challenging hypothesis is much too complex, as our assumption, and cannot be addressed completely with just one exploration. We invite the reviewer who are experienced in modelling including non-polynomial models, and we anticipate possible methodological collaboration worldwide and even from the anonymous reviewer for better modelling results in the future. We emphasize this point in the discussion.(P15, L494)

Reviewer 3 Report (Previous Reviewer 3)

Although the author did not provide a 1:1 response and I was tired of looking for it. But my comments have been revised.

Author Response

Point 1: Although the author did not provide a 1:1 response and I was tired of looking for it. But my comments have been revised.

Response 1: Thank the reviewer very much for the patience, dedication and insightful comments. We are really sorry for not providing a 1:1 response regularly.

As to the comments in the first-round reviewing, we would like to provide the point-by-point answer once more for reference as follows. Please see the attachment.

This manuscript is a resubmission of an earlier submission. The following is a list of the peer review reports and author responses from that submission.

Round 1

Reviewer 1 Report

This study analyzed data from 214 nations and regions to explore modelling and predicting of minimal CFR of emerging infectious disease during initial outbreak of COVID-19 pandemic. And this study presented a novel method to predict the minimal CFR, which shed light on quantitative support to establish suitable responding strategies worldwide, especially for nations and regions without death case in the initial dozens of days.

1. The references are inadequate. Please add recent 2 years research documents on case fatality rate.

2.Is CFR2 model available to evaluate Omicron epidemic?

Reviewer 2 Report

I am really sorry, but I do not think the proposed work can be published in its current form. The analysis is simple and not novel. The results are already obtained in previous research. The meta-analysis is almost not exist. The related-work is short and fails to capture the wide range of works in the field. Overall, I encourage the authors to check out the recent developments and try to adopt their work accordingly. My main comments are provided below:

Major:

1. The text is poorly written grammatically-wise. Almost every sentence is either bad-worded or grammatically wrong. I do not feel it can be published without massive proofreading - otherwise, it is super hard to understand anything.

2. Section 2.1 -  I am puzzled for the short duration declared in the paper. We are >700 days since the beginning of the pandemic so why only 113 is the longest duration?

3. Section 2.2 - you provide a lot of roles without any motivations. In this current form, it feels like cherry-picking roles.

4. Section 2.3. - is it a method or a result? You mixed the two...

5. I was just confused by Section 2.7. - I am not sure why we need it. How does it help in the analysis?

6. Section 3.1. - you dropped 122 out of 214 samples... this is super strange, you should provide a really good explanation for such a decision.

7. M1 & M2 in section 3.5. - did you try other, non-polynomial, models? I would suggest running symbolic regression to be sure this is the best function family to work with...

8. The entire work neglects the fact that recorded cases are under-sampled. The authors should address it somewhere in the text.

9.  Section 4 is really generic and I was not able to find any applicative recommendation, novel idea proposed based on the results, or some interesting analysis that combines the different results. It is not a discussion but a summary in the best case.

10. Section 5 - the claim is fair if only the authors show how one can extrapolate their results to new states or times frames. Lacking doing so, the authors can not propose the claim in section 5. 

Minor:

1. "To evaluate prediction efficiency of two established models (M1 and M2)" - at this point I am not sure what M1 and M2 are.

Reviewer 3 Report

I am lacking knowledge about such studies. I offer some comments for authors and editors according to my ideas:

1. line61, the authors are suggesting that "Some studies focused on the precise evaluation of COVID-19 disease severity with CFR" but there is only one literature. Could the authors list more?

2. I think the authors in chapter 2 only need to present the data sources and the applicable analysis methods, not to make the analysis. The authors can do the analysis and present the results in the RESULT section in Chapter 3.

3. Based on the results of the study, can the authors give some practical suggestions?